# WHY CONVOLUTIONAL NETWORKS LEARN ORIENTED BANDPASS FILTERS: A HYPOTHESIS

## ABSTRACT

It has been repeatedly observed that convolutional architectures when applied to image understanding tasks learn oriented bandpass filters. A standard explanation of this result is that these filters reflect the structure of the images that they have been exposed to during training: Natural images typically are locally composed of oriented contours at various scales and oriented bandpass filters are matched to such structure. The present paper offers an alternative explanation based not on the structure of images, but rather on the structure of convolutional architectures. In particular, complex exponentials are the eigenfunctions of convolution. These eigenfunctions are defined globally; however, convolutional architectures operate locally. To enforce locality, one can apply a windowing function to the eigenfunctions, which leads to oriented bandpass filters as the natural operators to be learned with convolutional architectures. From a representational point of view, these filters allow for a local systematic way to characterize and operate on an image or other signal.

## 1 INTRODUCTION

### 1.1 MOTIVATION

Convolutional networks (ConvNets) in conjunction with deep learning have shown state-of-the-art performance in application to computer vision, ranging across both classification, e.g., (Krizhevsky et al., 2012; Tran et al., 2015; Ge et al., 2019) and regression, e.g., (Szegedy et al., 2013; Eigen & Fergus, 2015; Zhou et al., 2017) tasks. However, understanding of how these systems achieve their remarkable results lags behind their performance. This state of affairs is unsatisfying not only from a scientific point of view, but also from an applications point of view. As these systems move beyond the lab into real-world applications better theoretical understanding can help establish performance bounds and increase confidence in deployment.

Visualization studies of filters that have been learned during training have been one of the key tools marshaled to lend insight into the internal representations maintained by ConvNets in application to computer vision, e.g., (Zeiler & Fergus, 2014; Yosinski et al., 2015; Mahendran & Vedaldi, 2015; Shang et al., 2016; Feichtenhofer et al., 2018). Here, an interesting repeated observation is that early layers in the studied networks tend to learn oriented bandpass filters, both in two image spatial dimenstions, $(x, y)^\top$, in application to single image analysis as well as in three spatiotemporal dimensions, $(x, y, t)^\top$, in application to video. An example is shown in Figure 1. Emergence of such filters seems reasonable, because local orientation captures the first-order correlation structure of the data, which provides a reasonable building block for inferring more complex structure (e.g., local measurements of oriented structure can be assembled into intersections to capture corner structure, etc.). Notably, however, more rigorous analyses of exactly why oriented bandpass filters might be learned has been limited. This state of affairs motivates the current paper in its argument that the analytic structure of ConvNets constrains them to learn oriented bandpass filters.

### 1.2 RELATED RESEARCH

Visualization of receptive field profiles (i.e., pointspread functions (Lim, 1990)) of the convolutional filters learned by contemporary ConvNets is a popular tool for providing insight into the image properties that are being represented by a network. A notable trend across these studies is that early layers

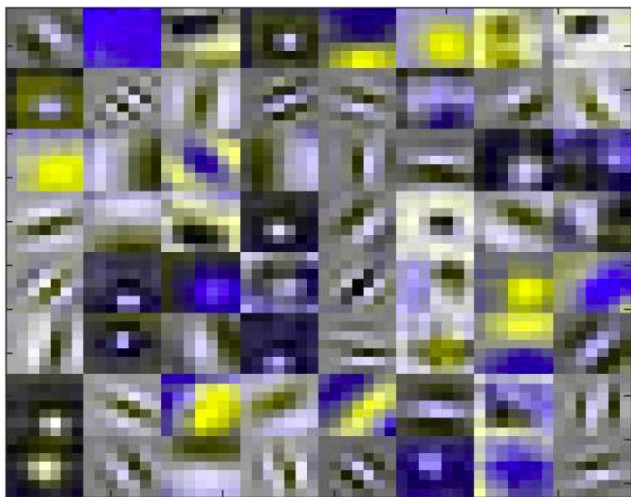

Figure 1: Visualization of pointspread functions (convolutional kernels) observed to be learned in the early layers of ConvNets. Brightness corresponds to pointwise function values. The majority of the plots show characteristics of oriented bandpass filters in two spatial dimensions, i.e., oscillating values along one direction, while remaining relatively constant in the orthogonal direction, even as there is an overall amplitude fall-off with distance from the center. The specific examples derive from the early layers of a ResNet-50 architecture (He et al., 2016) trained on ImageNet (Russakovsky et al., 2015).

appear to learn oriented bandpass filters in both two spatial dimensions, e.g., (Zeiler & Fergus, 2014; Springenberg et al., 2015; Yosinski et al., 2015; Shang et al., 2016) as well as three spatiotemporal dimensions, e.g., (Feichtenhofer et al., 2018). Indeed, earlier studies with architectures that also constrained their filters to be convolutional in nature, albeit using a Hebbian learning strategy (MacKay, 2003) rather than the currently dominant back-propagation approach (Rumelhart et al., 1986), also yielded filters that visualized as having oriented bandpass filter characteristics (Linsker, 1986). Interestingly, biological vision systems also are known to show the presence of oriented bandpass filters at their earlier layers of processing in visual cortex; see Hubel & Wiesel (1962) for pioneering work along these lines and for more general review DeValois & DeValois (1988).

The presence of oriented bandpass filters in biological systems often has been attributed to their being well matched to the statistics of natural images (Field, 1987; Olshausen & Field, 1996; Karklin & Lewicki, 2009; Simoncelli & Olshausen, 2001), e.g., the dominance of oriented contours at multiple scales. Similar arguments have been made regarding why such filters are learned by ConvNets. Significantly, however, studies have shown that even when trained with images comprised of random noise patterns, convolutional architectures still learn oriented bandpass filters (Linsker, 1986). These later results suggest that the emergence of such filter tunings cannot be solely attributed to systems being driven to learn filters that were matched to their training data.

Interestingly, some recent multilayer convolutional architectures have specified their earliest layers to have oriented bandpass characteristics (Bruna & Mallat, 2013; Jacobsen et al., 2016; Hadji & Wildes, 2017); indeed, some have specified such filters across all layers (Bruna & Mallat, 2013; Hadji & Wildes, 2017). These design decisions have been variously motivated in terms of being well matched to primitive image structure (Hadji & Wildes, 2017) or providing useful building blocks for learning higher-order structures (Jacobsen et al., 2016) and capturing invariances (Bruna & Mallat, 2013). Other work has noted that purely mathematical considerations show that ConNets are well suited to designs that capture multiscale, windowed spectra Bruna et al. (2016); however,it did not explictly established the relationship to eigenfunctions of convolution nor offered an explanation for why deep-learning yields oriented bandpass filters when applied to ConvNets.

In the light of previous research, the present work appears to be the first to offer an explanation of why ConvNets learn oriented bandpass filters by appeal to the inherent properties of their architectures. By definition, the convolutional layers of a ConvNet are governed by the properties of

convolution. For present purposes, a key property is that the eigenfunctions of convolution are complex exponentials. Imposing locality on the eigenfunctions leads to oriented bandpass filters, which therefore are the appropriate filters to be learned by a ConvNet.

## 2 ANALYTIC APPROACH

This section details a novel explanation for why ConvNets learn oriented bandpass filters. The first two subsections largely review standard material regarding linear systems theory (Oppenheim et al., 1983) and related topics (Kaiser, 2011; Kusse & Westwig, 2006), but are necessary to motivate properly our explanation. The final subsection places the material in the specific context of ConvNets.

### 2.1 EIGENFUNCTIONS OF CONVOLUTION

Let $\mathcal{L}$ be a linear operator on a function space. The set of eigenfunctions $\phi_n$ associated with this operator satisfy the condition (Kusse & Westwig, 2006)

$$\mathcal{L}\phi_n = \lambda_n \phi_n. \tag{1}$$

That is, the operator acts on the eigenfunctions simply via multiplication with a constant, $\lambda_n$, referred to as the eigenvalue. It sometimes also is useful to introduce a (positive definite) weighting function, $w$, which leads to the corresponding constraint

$$\mathcal{L}\phi_n = \lambda_n w \phi_n. \tag{2}$$

For cases where any function in the space can be expanded as a linear sum of the eigenfunctions, it is said that the collection of eigenfunctions form a complete set. Such a set provides a convenient and canonical spanning representation.

Let $\mathbf{x} = (x_1, x_2, \ldots, x_n)^\top$, $\mathbf{a} = (a_1, a_2, \ldots, a_n)^\top$ and $\mathbf{u} = (u_1, u_2, \ldots, u_n)^\top$. For the space of convolutions, with the convolution of two functions, $f(\mathbf{x})$ and $h(\mathbf{x})$ defined as

$$f(\mathbf{x}) * h(\mathbf{x}) = \int_{-\infty}^{\infty} f(\mathbf{x} - \mathbf{a}) h(\mathbf{a}) \, d\mathbf{a} \tag{3}$$

it is well known that functions of the form $f(\mathbf{x}) = e^{i\mathbf{u}^\top \mathbf{x}}$ are eigenfunctions of convolution (Oppenheim et al., 1983), i.e.,

$$\int_{-\infty}^{\infty} e^{i\mathbf{u}^\top (\mathbf{x} - \mathbf{a})} h(\mathbf{a}) \, d\mathbf{a} = e^{i\mathbf{u}^\top \mathbf{x}} \int_{-\infty}^{\infty} e^{-i\mathbf{u}^\top \mathbf{a}} h(\mathbf{a}) \, d\mathbf{a} \tag{4}$$

with the equality achieved via appealing to $e^{i\mathbf{u}^\top (\mathbf{x} - \mathbf{a})} = e^{i\mathbf{u}^\top \mathbf{x}} e^{-i\mathbf{u}^\top \mathbf{a}}$ and subsequently factoring $e^{i\mathbf{u}^\top \mathbf{x}}$ outside the integral as it is independent of $\mathbf{a}$. The integral on the right hand side of (4),

$$\int_{-\infty}^{\infty} e^{-i\mathbf{u}^\top \mathbf{a}} h(\mathbf{a}) \, d\mathbf{a}, \tag{5}$$

is the eigenvalue, referred to as the modulation transfer function (MTF) in signal processing (Oppenheim et al., 1983). Noting that $e^{i\mathbf{u}^\top \mathbf{x}} = \cos(\mathbf{u}^\top \mathbf{x}) + i \sin(\mathbf{u}^\top \mathbf{x})$ leads to the standard interpretation of $\mathbf{u}$ in terms of frequency of the function (e.g., input signal).

Given the eigenfunctions of convolution are parameterized in terms of their frequencies, it is useful to appeal to the Fourier transform of function $f(\mathbf{x})$, where we use the form (Horn, 1986)

$$\mathcal{F}(\mathbf{u}) = \int_{-\infty}^{\infty} f(\mathbf{x}) e^{-i\mathbf{u}^\top \mathbf{x}} \, d\mathbf{x}, \tag{6}$$

because any convolution can be represented in terms of how it operates via simple multiplication of the eigenvectors, (5), with the eigenfunctions, $e^{i\mathbf{u}^\top \mathbf{x}}$, with $\mathbf{u}$ given by (6). Thus, this decomposition provides a canonical way to decompose $f(\mathbf{x})$ and explicate how a convolution operates on it.

## 2.2 Imposing locality

Understanding convolution purely in terms of its eigenvectors and eigenvalues provides only a global representation of operations, as notions of signal locality, $\mathbf{x}$, are lost in the global transformation to the frequency domain, $\mathbf{u}$. This state of affairs often is unsatisfactory from a representational point of view because one wants to understand the structure of the signal (e.g., an image) on a more local basis (e.g., one wants to detect objects as well as their image coordinates). This limitation can be ameliorated by defining a windowed Fourier transform (Kaiser, 2011), as follows (Jahne & Hausbecker, 2000).

Let $w(\mathbf{x})$ be a windowing function that is positive valued, symmetric and monotonically decreasing from its center so as to provide greatest emphasis at its center. A Windowed Fourier Transform (WFT) of $f(\mathbf{x})$ can then be defined as

$$\mathcal{F}(\mathbf{u}_c, \mathbf{x}; w) = \int_{-\infty}^{\infty} f(\mathbf{a}) w(\mathbf{a} - \mathbf{x}) e^{-i\mathbf{u}_c^\top \mathbf{a}} \, d\mathbf{a}. \tag{7}$$

Making use of the symmetry constraint that we have enforced on the windowing function allows for $w(\mathbf{x}) = w(-\mathbf{x})$ so that the WFT, (7), can be rewritten as

$$\mathcal{F}(\mathbf{u}_c, \mathbf{x}; w) = \int_{-\infty}^{\infty} f(\mathbf{a}) w(\mathbf{x} - \mathbf{a}) e^{i\mathbf{u}_c^\top (\mathbf{x} - \mathbf{a})} e^{-i\mathbf{u}_c^\top \mathbf{x}} \, d\mathbf{a}, \tag{8}$$

which has the form of a convolution

$$f(\mathbf{x}) * \left( w(\mathbf{x}) e^{i\mathbf{u}_c^\top \mathbf{x}} \right) \tag{9}$$

with the inclusion of an additional phase component, $e^{-i\mathbf{u}_c^\top \mathbf{x}}$.

To provide additional insight into the impact the WFT convolution, (9), has on the function, $f(\mathbf{x})$, it is useful to examine the pointspread function, $w(\mathbf{x}) e^{i\mathbf{u}_c^\top \mathbf{x}}$, in the frequency domain by taking its Fourier transform (6), i.e., calculate its MTF. We have

$$\int_{-\infty}^{\infty} w(\mathbf{x}) e^{i\mathbf{u}_c^\top \mathbf{x}} e^{-i\mathbf{u}^\top \mathbf{x}} \, d\mathbf{x}, \tag{10}$$

which via grouping by coefficients of $\mathbf{x}$ becomes

$$\int_{-\infty}^{\infty} w(\mathbf{x}) e^{-i(\mathbf{u} - \mathbf{u}_c)^\top \mathbf{x}} \, d\mathbf{x}. \tag{11}$$

Examination of (11) reveals that it is exactly the Fourier transform of the window function, cf. (6), as shifted to the center frequencies, $\mathbf{u}_c$. Thus, operation of the WFT convolution, (9), on a function, $f(\mathbf{x})$, passes central frequency, $\mathbf{u}_c$, relatively unattenuated, while it suppresses those that are further away from the central frequency according to the shape of the window function, $w(\mathbf{x})$, i.e., it operates as a bandpass filter. Thus, convolution with a bank of such filters with varying central frequencies, $\mathbf{u}_c$, has exactly the desired result of providing localized measures of the frequency content of the function $f(\mathbf{x})$.

Returning to the pointspread function itself, $w(\mathbf{x}) e^{i\mathbf{u}_c^\top \mathbf{x}}$, and recalling that $e^{i\mathbf{u}^\top \mathbf{x}} = \cos(\mathbf{u}^\top \mathbf{x}) + i\sin(\mathbf{u}_c^\top \mathbf{x})$, it is seen that in the signal domain, $\mathbf{x}$, the filter will oscillate along the direction of $\mathbf{u}_c$ while remaining relatively constant in the orthogonal direction, even as there is an overall amplitude fall-off with distance from the center according to the shape of $w(\mathbf{x})$, i.e., we have an oriented bandpass filter.

As a specific example (Jahne & Hausbecker, 2000), taking $w(\mathbf{x})$ to be an n-dimensional Gaussian-like function, $g(\mathbf{x}; \sigma) = \kappa e^{-\|\mathbf{x}\|^2 / \sigma^2}$, with $\sigma$ the standard deviation and $\kappa$ a scaling factor, yields an n-dimensional Gabor-like filter,

$$g(\mathbf{x}; \sigma) e^{i\mathbf{u}_c^\top \mathbf{x}} = g(\mathbf{x}; \sigma) \left( \cos(\mathbf{u}_c^\top \mathbf{x}) + i\sin(\mathbf{u}_c^\top \mathbf{x}) \right), \tag{12}$$

which provides good joint localization of signal content in the signal and frequency domains Gabor (1946). Indeed, visualization of these filters in two spatial dimensions (Figure 2) provides strikingly similar appearance to those presented in Figure 1, if in an idealized form. In particular, their

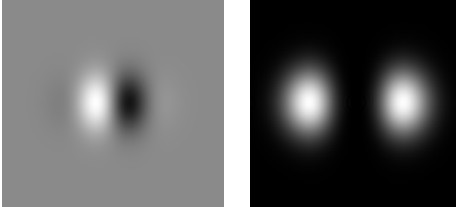

Figure 2: Visualization of an analytically defined oriented bandpass filter (12). The left panel shows the pointspread function corresponding to the odd symmetry (sin) component, while the right panel shows its power in the frequency domain. Brightness corresponds to pointwise function values.

pointspread functions oscillate according to a frequency $\|\mathbf{u}_c\|$ along the direction, $\frac{\mathbf{u}_c}{\|\mathbf{u}_c\|}$, while remaining relatively constant in the orthogonal direction, even as there is an overall amplitude fall-off with distance from the center. In the frequency domain, they have peak power at $\mathbf{u}_c$ with a fall-off following a Gaussian-like shape with standard deviation, $1/\sigma$, that is the inverse of that used in specifying the window, $w(\mathbf{x})$. These observations hold because we already have seen, (11), that the frequency domain representation of such a function is the Fourier transform of the window function, $w(\mathbf{x})$, shifted to the center frequencies, $\mathbf{u}_c$; furthermore, the Fourier tranform of a function of the form $g(\mathbf{x}; \sigma)$ has a similar form, albeit with an inverse standard deviation Bracewell (1986).

## 2.3 IMPLICATIONS FOR CONVNETS

Convolutions in ConvNets serve to filter the input signal to highlight its features according to the learned pointspread functions (convolutional kernels). Thus, convolution with the oriented filters shown in Figure 1 will serve to highlight aspects of an image that are correspondingly oriented and at corresponding scales. The question at hand is, "Why did the ConvNet learn such filters?" The previous parts of this section have reviewed the fact that complex exponentials of the form $e^{i\mathbf{u}^\top \mathbf{x}} = \cos(\mathbf{u}^\top \mathbf{x}) + i \sin(\mathbf{u}^\top \mathbf{x})$ are the eigenfunctions of convolution. Thus, such frequency dependent functions similarly serve as the eigenfunctions of the convolutional operations in ConvNets. In particular, this result is a basic property of the convolutional nature of the architecture, independent of the input to the system. Thus, for any convolution in a ConvNet the frequency dependent eigenfunctions, $e^{i\mathbf{u}^\top \mathbf{x}}$, provide a systematic way to represent their input.

As with the general discussion of locality presented in Subsection 2.2, for the specifics of ConvNets it also is of interest to be able to characterize and operate locally on a signal. At the level of convolution, such processing is realized via pointspread functions that operate as bandpass filters, (9). Like any practical system, ConvNets will not capture a continuous range of bandpass characteristics, as given by $\mathbf{u}_c$ and the sampling will be limited by the number of filters the designer allows at each layer, i.e., as a metaparameter of the system. Nevertheless, making use of these filters provides a systematic approach to representing the input signal.

Overall, the very convolutional nature of ConvNets inherently constrain and even define the filters that they learn, independent of their input or training. In particular, learning bandpass filters provides a canonical way to represent and operate on their input, as these serve as the localized eigenfunctions of convolution. As a ConvNet is exposed to more and more training data, its representation is optimized by spanning as much of the data as it can. Within the realm of convolution, in which ConvNet conv layers are defined, oriented bandpass filters provide the solution. They arise as the locality constrained eigenfunctions of convolution and thereby have potential to provide a span of any input signal in a localized manner. Thus, ConvNets are optimized by learning exactly such filters. Notably, since this explanation for why ConvNets learn oriented bandpass filters is independent of training data, it can explain why such filters emerge even when the training data lacks such pattern structure, including training on random input signals, e.g., (Linsker, 1986).

Two final implications suggest directions for future research. First, the analysis presented in this paper has been targeted toward an explanation for why ConvNets learn oriented bandpass filters in their early layers, as suggested in visualization studies discussed in Sec. 1.2. However, our analysis of oriented bandpass filtes as the localized eigenfunctions of convolution is not specific to early

ConvNet layers, but rather applies to any convolutional layer. The result thereby raises the question of whether deeper ConvNet layers also learn oriented bandpass filters. Here, empirical data is lacking as visualization studies to date for any given layer concentrate on the combined result across all previous layers, rather than the filtering characteristics at just that layer. Thus, interesting future research in examining the filters at each layer in isolation from those at other layers is motivated by the analysis presented in the current paper. Along these lines, it is interesting to note that certain hand-crafted ConvNets already make use of the same set of oriented bandpass filters at all layers of the architecture and do so to good advantage in their targeted tasks of single image and video texture analysis (Bruna & Mallat, 2013; Hadji & Wildes, 2017).

A second direction for future work involves use of parameterized filters during learning. In particular, the presented theoretical motivation for learning of oriented bandpass operators as the convolutional filters in ConvNets suggests the possibility of constraining the filters to be of that form in a learning-based framework. In such a framework, it would not be necessary for the training process to learn the numerical values for each and every individual filter value (i.e., each filter tap), but rather would merely need to learn a much smaller number of parameters, e.g., the values of the center frequency, $\mathbf{u}_c$ and the standard deviation $\sigma$ associated with the Gabor-like filter derived above (12). Such a constrained learning approach would require a much less intensive training procedure (e.g. involving far less data) compared to learnig values fo all individual filter taps, owing to the drastically reduced number of parameters that need to be estimated, even while being able to tune to the specifics of the task that is being optimized.

## 3 SUMMARY

Previous studies have demonstated that learned filters at the early layers of convolutional networks visualize as oriented bandpass filters. This phenomenon typically is explained via appeal to natural image statistics, i.e., natural images are dominated by oriented contours manifest across a variety of scales and oriented bandpass filters are well matched to such structure. We have offered an alternative explanation in terms of the structure of convolutional networks themselves. Given that their convolutional layers necessarily operate within the space of convolutions, learning oriented bandpass filters provides the system with the potential to span possible input, even while preserving a notion of locality in the signal domain.

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
