# OpenReview forum: "Why Convolutional Networks Learn Oriented Bandpass Filters: A Hypothesis"
_ICLR.cc/2020/Conference — Reject_

### Official Review · AnonReviewer2 · 2019-10-24
**Official Blind Review #2**

**Rating:** 3

**Review:**

This short, interesting paper provides a theoretical analysis to explain why we may expect to see bandpass oriented filters arise as a result of the convolutional structure of deep networks. The explanation boils down to the fact that the eigenfunctions of convolutions correspond to bandpass filters. This can explain surprising phenomena: bandpass filters arising when training on random noise inputs. The paper could have been even more interesting if some or more of the other questions had been considered:

1. The effect of training on a specific dataset had been explained: how do the resulting filters change and how does that tie in with the theory?

2. Some light shed on 3D filters in higher layers of the network.

3. Whether any guiding principles emerge for conv. layer construction.


**Experience Assessment:**

I have published in this field for several years.

**Review Assessment: Checking Correctness Of Derivations And Theory:**

N/A

**Review Assessment: Checking Correctness Of Experiments:**

N/A

**Review Assessment: Thoroughness In Paper Reading:**

N/A

---

### Official Review · AnonReviewer3 · 2019-10-25
**Official Blind Review #3**

**Rating:** 3

**Review:**

Research Problem: Previous studies have showed that learned filters at the early layers of CNNs visualize as oriented bandpass filters. This phenomenon typically is explained via appeal to natural image statistics, i.e., natural images are dominated by oriented contours manifest across a variety of scales and oriented bandpass filters are well matched to such structure.

Contribution: This paper proposes an explanation in terms of the structure of convolutional networks themselves: Given that
their convolutional layers necessarily operate within the space of convolutions, learning oriented bandpass filters provides the system with the potential to span possible input, even while preserving a notion of locality in the signal domain.

Question: What is possible implication and applications of this  explanation in practice. It will be useful to add discussion and experimental results to show the advantage of this explanation in real applications.

**Experience Assessment:**

I do not know much about this area.

**Review Assessment: Checking Correctness Of Derivations And Theory:**

I did not assess the derivations or theory.

**Review Assessment: Checking Correctness Of Experiments:**

N/A

**Review Assessment: Thoroughness In Paper Reading:**

I made a quick assessment of this paper.

---

### Official Review · AnonReviewer4 · 2019-10-31
**Official Blind Review #4**

**Rating:** 1

**Review:**

This paper claims that convolutional filters in CNNs are not the result of fitting to the input data distribution but they are the optimal solution to a spectral decomposition of the convolutional operator.

Positive things about this work:
1) it is clearly written
2) the fact that Gabor wavelets are the eigenfunctions of convolution is sound.
3) it provides good food for thought about what needs to be learned and what comes from the pre- specified choice of architecture

Negative things about this work:
1) it misses references to relevant works. In particular, there is an article making essentially the same points:
Joan Bruna, Soumith Chintala, Yann LeCun, Serkan Piantino, Arthur Szlam, and Mark Tygert, "A mathematical motivation for complex-valued convolutional networks," Neural Computation, 28 (5): 815-825, 2016
http://tygert.com/ccnet.pdf
where these authors make the same conclusions and observe that learning reduces to figuring out the  windowing, number of scales, etc. but not the type of filters.
This prior work greatly reduces the impact of this contribution, unfortunately.
There are other references that are missing, but these are minor points compared to the above. For instance, I'd recommend to cite Hubel & Wiesel's seminal work on mapping the mammalian receptive fields, and older work by M. Lewicki about analyzing learned receptive fields by sparse coding algorithms, similarly to the cited B. Olshausen et al.
2) The Authors show that Gabor wavelets are eigenfunction of convolutions and they stop here to conclude that filters in CNNs are the way they are because of the architecture, but how about the effect of the non-linearities, depth and the type of cost used for training? The work is unfinished without a thorough analysis and discussion of these crucial aspects.

**Experience Assessment:**

I have published in this field for several years.

**Review Assessment: Checking Correctness Of Derivations And Theory:**

I assessed the sensibility of the derivations and theory.

**Review Assessment: Checking Correctness Of Experiments:**

N/A

**Review Assessment: Thoroughness In Paper Reading:**

I read the paper at least twice and used my best judgement in assessing the paper.

---

### Official Review · AnonReviewer1 · 2019-11-01
**Official Blind Review #1**

**Rating:** 3

**Review:**

This paper proposed a hypothesis on why neural network learns oriented bandpass filters. While most existing work attribute this phenomenon to image structures, this paper suggests that it might be a property of convolution. In particular, it shows Fourier basis are eigenfunctions of convolution, and band pass filters are eigenfunctions for the generalized eigen problem of convolution given a windowed weighting function, which corresponds to a windowed Fourier transform.

The mathematical observations are interesting, and the paper hypothesizes that this mathematical property encourages neural networks to learn oriented bandpass filters. However, it is unclear why the neural network should learn eigenfunctions as the filters. I understand the paper is proposing a hypothesis, but drawing a more solid conclusion is important. I am not recommending acceptance of this paper in the main conference, but it may be a good paper in a certain workshop.

Besides, does higher layer of a deep neural network also learn bandpass filters w.r.t. its input feature map? How well the phenomenon and the hypothesis could generalize to the deeper layers?


**Experience Assessment:**

I have read many papers in this area.

**Review Assessment: Checking Correctness Of Derivations And Theory:**

I assessed the sensibility of the derivations and theory.

**Review Assessment: Checking Correctness Of Experiments:**

N/A

**Review Assessment: Thoroughness In Paper Reading:**

I read the paper at least twice and used my best judgement in assessing the paper.

---

### Decision · Program_Chairs · 2019-12-19

**Decision:**

Reject

**Comment:**

This paper proposes an alternative explanation of the emergence of oriented bandpass filters in convolutional networks: rather than reflecting observed structure in images, these filters would be a consequence of the convolutional architecture itself and its eigenfunctions.
Reviewers agree that the mathematical angle taken by the paper is interesting, however they also point out that crucial prior work making the same points exists, and that more thorough insights and analyses would be needed to make a more solid paper.
Given the closeness to prior work, we cannot recommend acceptance in this form.